# Smart Dairy Farming—The Potential of the Automatic Monitoring of Dairy Cows’ Behaviour Using a 360-Degree Camera

**DOI:** 10.3390/ani14040640

**Published:** 2024-02-16

**Authors:** Friederike Kurras, Martina Jakob

**Affiliations:** Department of Technological Assessment and Substance Cycles, Leibniz Institute for Agricultural Engineering and Bioeconomy (ATB), Max-Eyth-Allee 100, 14469 Potsdam, Germany; mjakob@atb-potsdam.de

**Keywords:** dairy behaviour, computer vision, precision livestock farming, animal welfare, lying behaviour

## Abstract

**Simple Summary:**

This study investigates the extent to which a 360° camera is suitable for detecting the behaviour of cows in a dairy barn. The focus is on animal behaviour, as this is a suitable indicator that allows conclusions about animal health. Therefore, 2299 snapshots from a 360° camera installed in a dairy cattle barn were evaluated manually to identify lying, standing and eating behaviours. Based on the principle of the multi-moment study, the number of animals in each image is determined according to the behavioural patterns. During the 18 h of observation, the average lying time per cow was 5.9 h, standing time was 6.9 h, and feeding time was 5.2 h. Since the accuracy rate of correctly detected animals is 93.1%, the camera technology seems to be a suitable tool for observing animal behaviour and, thus, their health in real time. In the ongoing project, the behavioural patterns will be recognised automatically using artificial intelligence.

**Abstract:**

The aim of this study is to show the potential of a vision-based system using a single 360° camera to describe the dairy cows’ behaviour in a free-stall barn with an automatic milking system. A total of 2299 snapshots were manually evaluated, counting the number of animals that were lying, standing and eating. The average capture rate of animals in the picture is 93.1% (counted animals/actual numbers of animals). In addition to determining the daily lying, standing and eating times, it is also possible to allocate animals to the individual functional areas so that anomalies such as prolonged standing in the cubicle or lying in the walkway can be detected at an early stage. When establishing a camera monitoring system in the future, attention should be paid to sufficient resolution of the camera during the night as well as the reduction of the concealment problem by animals and barn equipment. The automatic monitoring of animal behaviour with the help of 360° cameras can be a promising innovation in the dairy barn.

## 1. Introduction

Animal behaviour displays the physiological condition of an individual and is an important indicator for the evaluation of animal health and welfare [1]. For example, an increased locomotor activity can be an indicator of oestrus [2] or may indicate discomfort, lameness or other problems in health, management or social structure. Standing and lying behaviour has already been widely studied in the context of animal health [3].

The animal behaviour can be monitored by direct observation or by using a variety of sensors. While direct and continuous observations are associated with a high input of labour time, the use of sensors can ensure monitoring of the animals with relatively little effort. Sensors can basically be divided into ‘at cow’, ‘near cow’ and ‘from cow’ sensors [4]. The ‘at cow’ sensors range from acceleration sensors to sensors for measuring body temperature, rumination activity, heart rate or pH value and are designed to detect specific parameters or behaviours such as standing, lying, walking or chewing. The ‘at cow’ sensors have high precision [5] but are exposed to mechanical influences such as pressure, dirt and liquids. The ‘near cow’ sensors, such as cameras, microphones, climate sensors or GPS systems, are placed in the barn so that the animals walk past, over or through them. ‘From cow’ sensors monitor products derived from the animal, i.e., milk, hair or dung. These sensors often already include promising machine learning technologies. In recent years, a large number of studies have investigated a wide variety of use cases and methods for automatic detection of animal-specific parameters based on machine vision like lameness detection through gait analysis [6,7,8], drinking [9,10], eating [11,12] and chewing behaviour [10], as well as social interaction [13,14]. The use of a camera-based monitoring system in the barn could be a promising application, but it is still in its infancy, and only a few systems are commercially available so far. However, the basis for automatic behaviour detection is object recognition, in this case, of the individual animal, as well as its location and tracking. So far, these functions are still only realised for a limited number of animals, especially since tracking is difficult in larger herds [15].

The objectives of this research were to (1) describe the dairy cows’ behaviour in a free-stall barn with an automatic milking system and (2) show the potential of a single 360° camera as a low-budget vision-based application. In order to address these research objectives, we evaluated snapshots of a 360° fisheye camera over a duration of 14 days. This research is part of the smartMILC project. The aim of the project is to develop a holistic system for the evaluation of animal health and welfare on the basis of a digital twin [16].

## 2. Materials and Methods

The observation of animal behaviour took place in a freely ventilated open barn during operation in Brandenburg, Germany, from 9 to 23 June in 2023. The herd size fluctuated from 64 to 67 cows of Holstein Friesian breed during the study period due to animal removals (drying off, group changes and illness). The barn offers 46 feeding places and 68 lying cubicles, resulting in a 1:1 cubicle ratio for the animals. The cubicles are high boxes equipped with mattresses and littered with a straw–lime mixture. The walkway consists of slatted floor, which is cleaned by an automatic manure pusher. The barn has three drinkers, 6 fans and an automatic milking system. The animals are fed twice a day at 5.30 a.m. and 10 a.m. An automatic feed pusher ensures that the feed is regularly pushed up. The average milk yield of the group was 34.5 L/day/cow and 2.6 milkings per day/cow.

In February 2023, a 360° fisheye camera for a complete barn overview and four Intel RealSense depth cameras for selected areas were installed in the barn (Figure 1). For the purposes of this study, only the footage of the 360° camera was evaluated, as it already provides a good overview of the different activities of the animals in the barn.

In an initial investigation, parameters that can be automatically detected by computer vision have been identified [17]. During a direct observation on 20 April 2023, the practicability of some of these parameters was evaluated in the research barn. For this purpose, the number of cows eating, lying (in the cubicles and in the alley) and standing (in the alley or cubicles), as well as walking animals and the use of the automatic milking system, were counted every 10 min. The parameters and the monitoring frequency of 10 min turned out to be sufficient and were chosen for the subsequent study period. It continued recording for two weeks, resulting in 2299 snapshots, but 690 were discarded due to poor image resolution at night.

To obtain the temporal distribution of activities in the barn, we use the principle of a multi-moment study, in which the frequency of fixed processes is recorded by random short-term observations. In this case, the aforementioned parameters, number of cows lying (in the cubicle or walkway), standing (in the cubicle or walkway) and eating, were counted manually in snapshots recorded with the 360° camera (Figure 2).

In the first step, the number of necessary observations of the individual behaviour was determined (n = $\frac{1.96^{2}\times p\times (100\%-p)}{f^{\prime }²}$). Therefore, the required confidence interval f′ was set at 2.5, which means that the information has a confidence level of 95%; 1.96 is a statistical safety factor that is to be applied with a probability of 95%. Finally, the number of images to be analysed was calculated by dividing the number of necessary observations (n) by herd size (Table 1).

The evaluation of the frequency of the detected behaviour allows the calculation of the timely distribution of the defined behaviour within the study period and is calculated as follows:$$\mathrm{Calculation}\;\mathrm{of}\;\mathrm{the}\;\text{percentage:}\;P\;(\mathrm{behaviour})=\frac{Specific\;behaviour\times 100\%}{Sum\;observations\;per\;day}$$
$$\mathrm{Calculation}\;\mathrm{time}\;\mathrm{per}\;\mathrm{day}=\frac{Observed\;behaviour\left(\%\right)\times Sum\;observed\;time\;(min.)}{100\%}$$

## 3. Results

The basis for the following data analysis is 1609 snapshots. Only the time period from 4 a.m. to 10 p.m. was evaluated, as the quality of the night images does not allow a reliable count of animals. Thus, a complete assessment of the circadian rhythm is not possible from our set of data. Nevertheless, the animals in the barn showed a diurnal rhythm (Figure 3). Especially shortly before feeding time, many animals were waiting in the walkway of the feeding table. When the feed is presented, the number of standing and lying animals decreases. Between 8 a.m. and 7 p.m., there is little fluctuation in the behavioural patterns. As the second feeding takes place 4 h after the first feed presentation, the animals are less motivated to go to the feed table again. Around 7 p.m., the number of eating animals reaches another peak but then decreases again.

During an observation period of 18 h per day, the average lying time per cow was 5.9 h, the average standing time was 6.9 h, and the average feeding time was 5.2 h (Figure 4).

Standing time was differentiated according to the functional areas because this gives an insight into possible health or barn hygiene issues. (Figure 5).

The average detection rate of counted animals (counted animals/actual numbers of animals) was 93.1%. The detection rate was lower in the morning and evening hours due to the decreasing image quality caused by the lighting conditions.

## 4. Discussion

An average detection rate of 93.1% (correctly counted number of animals) is a promising result and should encourage researchers to proceed with developing an automatic system based on one single low-budget camera. To ensure the reliable detection of animal behaviour, the camera must be selected in order to achieve sufficient accuracy even in adverse lighting conditions. In our sample, the resolution had a limiting effect on the image quality, and therefore, no observation was possible during the night [10]. The evaluation of the camera data was additionally disturbed by the concealment of the barn equipment hiding standing animals [18]. Therefore, a camera-based monitoring system might need more than one camera, depending on the barn size and its setup. Newly constructed barns should be planned according to these needs, such as having free vision of all areas.

As DeVries and von Keyserlingk [19] noted, the feeding behaviour of cows kept indoors is influenced in particular by the presentation of fresh feed. As the second feeding took place only 4 h after the first in our barn, the animals did not show a high motivation to go to the feed table again. Albright [20] found that the feed intake of cows is adapted to sunrise and sunset. The second peak of eating animals at 7 p.m. can be explained by this. From 9 p.m. to 10 p.m., the number of lying animals increases, so we expect a further increase during the night [3].

The daily lying time in this study is comparatively low [3]. The cows’ lying behaviour follows the light–dark rhythm, and therefore, we expected that the majority of animals lie during nighttime. Since the night data were not evaluated due to the low image quality, the lying times are underestimated, and the standing times are overestimated. No significant differences were found in the distribution of the observed behaviours between each day.

In the present study, standing time was differentiated according to the functional areas (Figure 3). The observed standing time in the cubicles is comparatively high. Studies have shown that animals standing in cubicles and in the alley can be an indication of a less-than-optimal barn design [21]. Prolonged standing time in the cubicle is observed especially if the quality of the cubicle is insufficient (e.g., too hard or too wet) or if the cubicle is too small [3]. Heat stress can also be a cause of extended standing times. Thus, in this experiment, an increase in standing animals was observed with increasing temperature (minimum: 16.25 °C, maximum: 29.34 °C). Besides rising temperatures, lameness may also be a cause for prolonged standing times. Cook et al. [22] found that non-lame animals increased standing in the walkway during heat stress to take advantage of the cooling effect of the fans. Meanwhile, lame cows preferred to stand in cubicles with soft surfaces. Camera-based detection of animal behaviour allows one to locate the animal in the functional area. Thus, anomalies such as lying in the walkway or standing in the cubicle can be detected, and management measures can be taken to increase animal welfare. In contrast, the attached sensors using accelerometers indicate the activities of a cow accurately [23], but they do not allow us to determine where the animals are located at any given time. Camera-based systems have a clear advantage in this case. We are expecting further information about animal behaviour once the camera data are analysed in more detail using artificial intelligence.

In summary, the automatic recording and evaluation of animal behaviour using a camera system is a promising application. The advantage of a video-based system is the simultaneous non-invasive study of the behaviour of multiple animals and different behavioural patterns at the same time through only one or a few devices. This is important, as increasing automation reduces contact between humans and animals and, at the same time, increases the possibility of detecting health problems in the herd through observation. In addition, the use of camera technology promises to be a cost-effective alternative since animal-based sensors are becoming obsolete due to camera technology. Nevertheless, problems such as occlusion by stable objects or animals, resolution of the image in adverse lighting conditions and soiling of the lens must be solved. For the farmer, such a monitoring system is a good way to improve animal welfare without a large additional daily workload. In the future, the behavioural data should be put into context with other sensor data (e.g., milking and weather data) so that a comprehensive overview of animal health can be provided and adverse effects such as heat and prolonged standing are put into context. However, the integration of different sensor data in one system is difficult to reconcile, among other things, due to manufacturer interests.

## 5. Conclusions

This study has shown that it is possible to detect specific behavioural patterns of individual animals in a dairy herd using a 360° camera. The use of artificial intelligence can enable real-time evaluation of animal behaviour, provide direct conclusions about animal health and inform the farmer immediately. In addition, the use of camera technology promises to be a cost-effective alternative, as it could replace animal-based sensors.

For the ongoing work, it is important to train algorithms that can identify the individual animal and automatically recognise behaviour such as lying, standing and walking.

## Figures and Tables

**Figure 1 animals-14-00640-f001:**
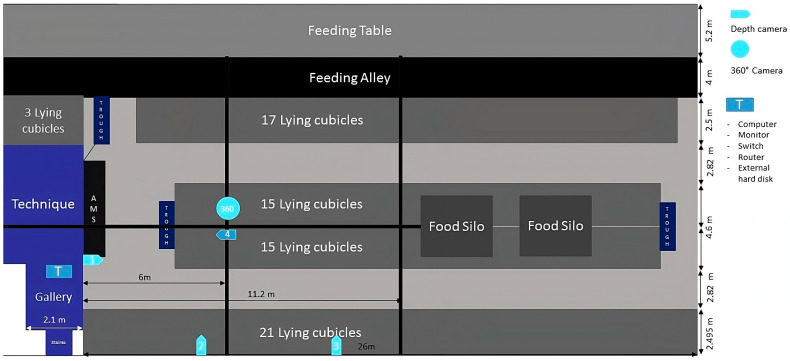
Research barn, technical installation.

**Figure 2 animals-14-00640-f002:**
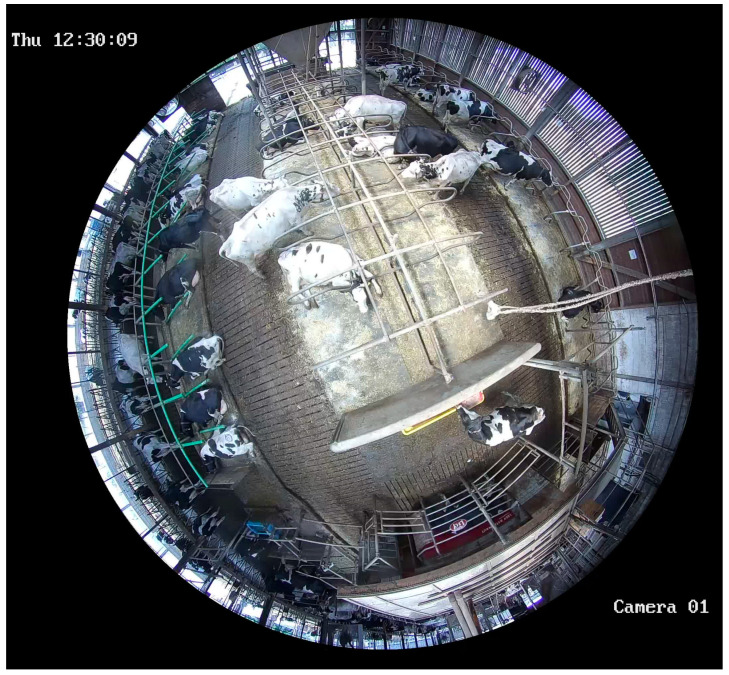
Snapshot taken by the 360° fisheye camera.

**Figure 3 animals-14-00640-f003:**
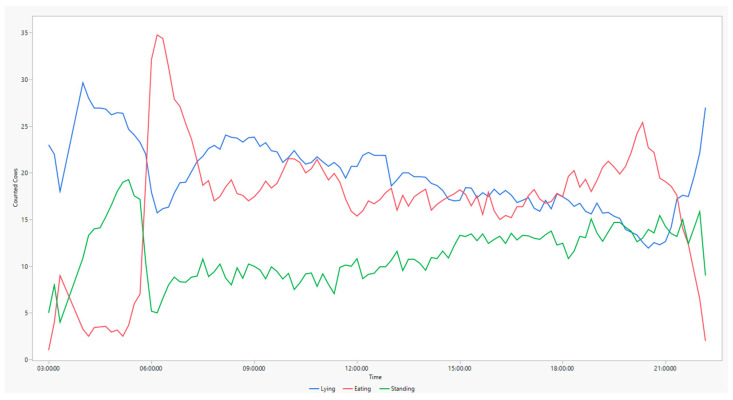
Number of cows lying, standing and eating depending on time of the day.

**Figure 4 animals-14-00640-f004:**
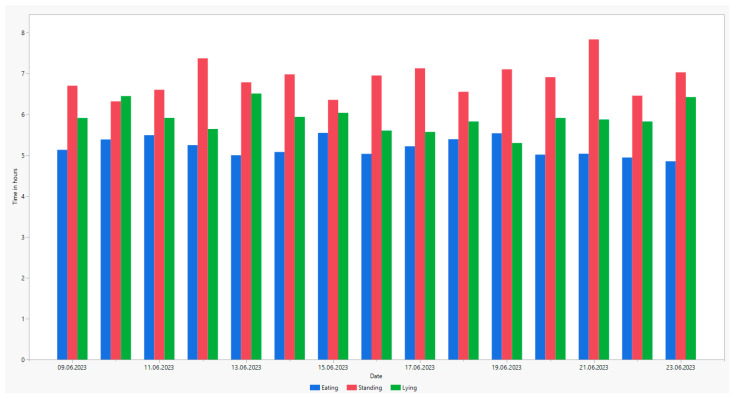
Average daily lying, standing and eating time during test period in hours.

**Figure 5 animals-14-00640-f005:**
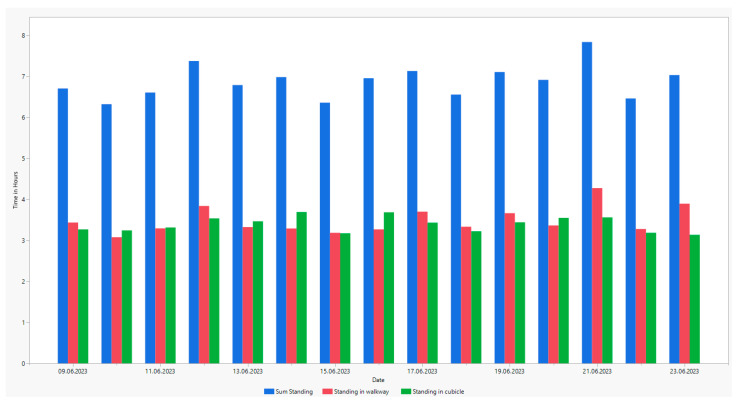
Comparison of standing times according to functional areas (sum of time standing, standing in walkway and standing in cubicle).

**Table 1 animals-14-00640-t001:** Statistical values for the multi-moment study.

Behaviour	Presumed Percentage of the Behaviour (p in %)	Absolute Confidence Interval (f′ in %)	Number of Required Observations (n)
Lying	50	2.5	1.537
Lying down process	0.5	2.5	31
Standing up process	0.5	2.5	31
Standing	23	2.5	1.089
Eating	25	2.5	1.153
Milking	1	2.5	61

## Data Availability

Data available on request due to restrictions (e.g., privacy, legal or ethical reasons).

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
