# Peer review of "Smart Dairy Farming—The Potential of the Automatic Monitoring of Dairy Cows’ Behaviour Using a 360-Degree Camera"

_animals, 2024, doi:10.3390/ani14040640_

Round 1

Reviewer 1 Report

Comments and Suggestions for Authors

This is an interesting article for someone who is embarking on the use of video analytics for determining animal behaviour because it give a good overview of the types of behaviour that are relevant. 

The data analysis I assume was made by using a human to look at the videos and make an assessment of parameters - eg number of cattle, number standing etc. This was not wholly clear and perhaps could be made more explicit.

The analysis of the number of observations that is required to be statistically significant is something that animal scientists are more familiar with than engineers typically. If the authors could expand on how they reached these numbers it might be of educational benefit. I would suggest that this is a useful thing to do given the fact that there is no real engineering content to the paper and the topic is really an application of engineering ie the potential for using video analytics.

Author Response

Dear Reviewer,

thank you very much for your comments and remarks on our article. Below you will find an overview with a response to the reviewers' comments as well as a changelog.

Best wishes,

Martina Jakob & Friederike Kurras

Comment

Answer

Change in Manuscript

Review 1

1

The data analysis I assume was made by using a human to look at the videos and make an assessment of parameters - eg number of cattle, number standing etc. This was not wholly clear and perhaps could be made more explicit.

Many thanks for the hint! We have described in the summary, abstract and materials section that this is a manual evaluation.

Lines: 11, 20, 94

2

The analysis of the number of observations that is required to be statistically significant is something that animal scientists are more familiar with than engineers typically. If the authors could expand on how they reached these numbers it might be of educational benefit.

Lines 99-112 describe how the minimum number of images to be evaluated was determined and how the daily lying, standing and feeding times were subsequently calculated. Adjustments were made for better understanding.

Lines 99-112

Reviewer 2 Report

Comments and Suggestions for Authors

1.- the abstract must contain the summarized results, preferably quantitative.

1.- The introduction shows a significant lack of studies with deep learning and monitoring of farm animals

2.- The quality of the images must be improved.

3.- Is animal identification carried out automatically? What algorithm was used?

4.- The use of the data obtained must be explored in depth; some examples of the use of this type of data.

5.- What were the errors due to? Don't discuss the identification weaknesses or using a camera with the high distortion caused by 360 degrees.

6.- The writing of the article must be corrected; It has many spelling and grammatical errors.

Comments on the Quality of English Language

The writing of the article must be corrected; It has many spelling and grammatical errors.

Author Response

Dear Reviewer,

thank you very much for your comments and remarks on our article. Below you will find an overview with a response to the reviewers' comments as well as a changelog.

Best wishes,

Martina Jakob & Friederike Kurras

Review 2

3

The abstract must contain the summarized results, preferably quantitative.

We have added quantitative information to the summary.

During the 18 hours of observation, the average lying time per cow was 5.9 hours, standing time was 6.9 hours and feeding time was 5.2 hours. Since the accuracy rate of correctly detected ani-mals is 93.1%, the camera technology seems to be a suitable tool for observing animal behaviour and thus their health in real time.

4

The introduction shows a significant lack of studies with deep learning and monitoring of farm animals

The introduction mentions systems that are already being used in practice. As this article focuses exclusively on the potential of a 360° camera for a complete overview of the barn, the articles and findings on monitoring barn animals already mentioned have not been expanded.

No changes in manuscript.

5

The quality of the images must be improved.

The images have been replaced and inserted in better quality.

Changed in manuscript.

6

Is animal identification carried out automatically? What algorithm was used?

We have described in the summary, abstract and materials section that this is a manual evaluation.

Lines: 11, 20, 94

7

The use of the data obtained must be explored in depth; some examples of the use of this type of data.

The data and results will be used for future work. This study investigates the feasibility of using a 360° camera to observe the entire herd.

Lines: 201-208

8

What were the errors due to? Don't discuss the identification weaknesses or using a camera with the high distortion caused by 360 degrees.

Errors have already been discussed in the manuscript. These include the poor resolution of the camera recordings during the night, which did not allow these images to be analyzed. In addition, the concealment by stable elements and animals posed a problem (lines 144-151).

No changes in manuscript.

9

The writing of the article must be corrected; It has many spelling and grammatical errors.

Corrected in manuscript.

Round 2

Reviewer 2 Report

Comments and Suggestions for Authors

The authors have addressed my main concerns.

Comments on the Quality of English Language

Minor editing of English language required